Application of a reaction-based water quality model to the total dissolved solids concentration of the Pasig River

Abas Crisanto L. 1 2 3 clabas@up.edu.ph
Velasco Arrianne Crystal 1 3
http://orcid.org/0000-0001-5649-9750 Arceo Carlene 1 3
1 Institute of Mathematics, University of the Philippines Diliman , Quezon City, Metro Manila , Philippines
2 Department of Mathematics, Visayas State University , Baybay City, Leyte , Philippines
3 Natural Sciences Research Institute, University of the Philippines Diliman , Quezon City, Metro Manila , Philippines
Scheibe Timothy
Electronic publication date: 2024 Oct 7
Publication date: 2024
Volume: 12
Electronic Location ID: e18076
Received 2024 Apr 18; Accepted 2024 Aug 20
Copyright: © 2024 Abas et al.
Copyright year: 2024
Copyright holder: Abas et al.
License: This is an open access article distributed under the terms of the Creative Commons Attribution License, which permits unrestricted use, distribution, reproduction and adaptation in any medium and for any purpose provided that it is properly attributed. For attribution, the original author(s), title, publication source (PeerJ) and either DOI or URL of the article must be cited.
License URL: https://creativecommons.org/licenses/by/4.0/

Keywords: Water quality model, Partial differential equation, Finite element method, Pasig River, Parameter estimation, Sensitivity analysis

Funding: Natural Sciences Research Institute University of the Philippines Diliman MAT-24-1-07 This work was supported by the Natural Sciences Research Institute, University of the Philippines Diliman, under Research Grant MAT-24-1-07. The funders had no role in study design, data collection and analysis, decision to publish, or preparation of the manuscript.

==============================
With the goal to support effective water resource management, water quality models have gained popularity as tools for evaluating the distributions of pollutants and sediments. This work focuses on the application of the numerical solution of an advection-dispersion-reaction (ADR) water quality model for rivers and streams to a major Philippine waterway, the Pasig River. The water quality constituent is described by a system of reaction and advection-dispersion-reaction equations. The model and method are based on a previously used strategy where Guass-Jordan decomposition is applied to the matrix system and the resulting conservative form of the model is solved numerically using the fully implicit scheme and finite element method. The methodology is demonstrated by a case study in Pasig River involving the concentrations of total dissolved solids (TDS) obtained from the Department of Environment and Natural Resources (DENR) through the Pasig River Unified Monitoring Stations (PRUMS) report. Sensitivity analysis and parameter estimation are also applied to the model to assess which parameters influence the model output the most.

Introduction

For centuries, people and their economy relied mostly on the conditions of the river system. It provides great benefits, such as clean water, irrigation, food sources, transportation, energy, and more, to the communities. The major factor that drives the success of the community within the river system is the quality of the water. The good quality of the river water, if managed properly, can boost the economy of not only the community but also the nation. It can also sustain fisheries, aquatic resources, and the ecology of rivers (Loucks & van Beek, 2017b).

Due to human activities, natural disasters, and/or climate change, river conditions change, which may have negative impacts on people and ecology (Loucks & van Beek, 2017b). By knowing more about these changes in conditions, people should be able to adapt or be better prepared to prevent disastrous effects on the community. One way to be prepared is to be able to foresee and assess the changes in the water quality of the river, then identify what causes the change (human activities, extreme weather conditions, flood, discharge, etc.,) and try to find a solution, if possible. According to a study from Utrecht University in the Netherlands, water management is facing significant problems from climate change, the increasing frequency of droughts, and rainstorms. Water quality is also under threat, in addition to its availability (van Vliet et al., 2023).

Mathematical modeling is particularly useful for assessing water quality. River water resources have been efficiently monitored and managed by implementing water quality models. Water quality specialists at the US Environmental Protection Agency utilize models for numerous goals (Tech et al., 2018). Evaluating water quality conditions and reasons for degradation.

Predicting how lakes and rivers will react to changes in their watersheds and surroundings (for example, future expansion, and climate change).

Estimating the quantitative advantages of new surface water protection policies.

In 2015, the United Nations established the Sustainable Development Goals (SDGs), commonly referred to as the Global Goals, as a global call to action to end poverty, safeguard the environment, and guarantee that by 2030, people will live in peace and prosperity. Studies on water quality models and their applications are aligned with the SDG. The alignment is with SDG 6 (Clean Water and Sanitation), which guarantees everyone has access to and sustainable management of water and sanitation, SDG 11 (Sustainable Cities and Communities), which creates inclusive, secure, resilient, and sustainable cities and human settlements, and SDG 13 (climate action), which emphasizes that initiatives to incorporate disaster risk reduction, sustainable natural resource management, and human security into national development must be coordinated with one another (UNDESA, 2023; UNDP, 2023).

Various water quality models can help forecast the water quality implications of different land and water management policies and practices (Loucks & van Beek, 2017a). Sediment transport is one of the water quality models mostly studied in river or stream networks (Wang et al., 2008). A study was conducted for sediment transport and hydrodynamic modeling in which the concentration of suspended sediments is being monitored with the effects of tides and waves near the estuary (Yang et al., 2022). Artificial neural networks (ANN) and machine learning techniques have been employed in more recent studies involving sediment transport to simulate flow and sediment transport in alluvial rivers (Ara Rahman & Chakrabarty, 2020; Roushangar, Shahnazi & Azamathulla, 2023; Chen et al., 2020). A survey of water quality modeling using artificial intelligence (AI) models has also been conducted where a number of AI models were used and applied in different river systems all around the world (Tiyasha, Tung & Yaseen, 2020).

Chemical transport is a major area of research in the studies of water quality models in rivers and streams (Park & Lee, 2002; Boorman, 2003; Lopes et al., 2004). In order to determine its potential for modeling water quality constituents in rivers, a comprehensive analysis of the most popular water quality models (DRAINMOD, ECM, MIKE-11, SIMCAT, QUAL2K, etc.) at the moment has been conducted (Cox, 2003; Tsakiris & Alexakis, 2012). One of the most popular water quality models used today is the QUAL2K. QUAL2K is a one-dimensional water quality model for rivers and streams that incorporates water-quality kinetics, non-uniform steady flow, and steady-state hydraulics (Chapra, Pelletier & Tao, 2012; Ahmad Kamal, Muhammad & Abdullah, 2020). QUAL2K can simulate a number of water-quality constituents (conductivity, suspended solids, phosphorous, etc.). Heavy metals in dissolved phase and other water quality parameters in rivers have been simulated and modeled using the advection-dispersion equation (Khalilzadeh Poshtegal & Mirbagheri, 2023; Paudel et al., 2022). The advection-dispersion equation model is also used in analyzing pollutant distribution in rivers and streams (Permanoon, Mazaheri & Amiri, 2022; Atshan et al., 2020; Mannina & Viviani, 2010). Nevertheless, the models for sediment and chemical transport that have been mentioned either model a particular system or are restricted to particular chemical species or reactions. Moreover, the mathematical formulation of these models mostly involves only ordinary differential equations with time as the independent variable. That is, they only identify how the water quality changes with respect to time on a specific position/element on the river and do not classify how the concentration changes as it goes along the river. While these models are useful, they may not be applicable in other environmental situations and are limited to being effective monitoring and management tools for the particular system for which they were designed and validated. Furthermore, according to Tsakiris & Alexakis (2012), many models created without financial resources are insufficiently user-friendly because of an expensive component of the software code for the proper user interface. As a result, these water quality models are rarely used and are typically expensive to obtain (Tsakiris & Alexakis, 2012).

With better knowledge and mathematical representations of numerous biogeochemical interactions (Thomann, 1998; SomlyóDy et al., 1998; Mann, 2000; Yeh, Burgos & Zachara, 2001), more broadly applicable generic models have been developed by Yeh et al. (2005) that can simulate user-prescribed reaction networks. Unlike the other models, the mathematical formulation here separates each factor that affects the concentration of the water quality constituent, e.g., advection term, dispersion term, and reaction term. In contrast, in others, the concentration is not explicitly affected by the change in distance, and the reaction term is incorporated in the sources and sinks. Furthermore, the formulation considers the complex biogeochemical interactions of the water quality constituents (Zhang et al., 2008).

A simple model considers only the advection-dispersion-reaction (ADR) of a single water quality constituent (Fernandes & Karney, 2001). The advantage of the model of Yeh et al.’s (2005) is that it does not only consider the ADR but also simultaneously looks at the interactions/reactions of the water quality constituents with other water quality constituents (Zhang et al., 2008).

In order to characterize a reactive system, Yeh et al. (2005) categorized each biogeochemical reaction as kinetic or equilibrium, especially in transport simulation of water quality parameters involved in these reactions (Rubin, 1983; Zhang et al., 2008). There are few reaction-based watershed models, e.g., QUAL2K, MIKE-11, that can address kinetic reactions in the transfer of chemicals and sediments (Yeh et al., 2005; Cheng, Yeh & Cheng, 2000; Zhang et al., 2008). A model for the equilibrium speciation, kinetic reaction, and transport of trace metals in saturated porous media with organic substrate biodegradation has been developed in Smith & Jaffe (1998). It is also important to emphasize that the kinetics of biogeochemical processes are affected by the critical role of transport in rivers or stream settings (Steefel, 2008).

In most cases, ordinary differential equations are used to formulate the model, e.g., Chapra, Pelletier & Tao (2012), Boorman (2003), Skaggs, Youssef & Chescheir (2012), and the direct method was employed in these watershed models, wherein the concentrations of water quality variables are obtained by directly integrating/solving the ordinary/partial differential equations (PDEs) regulating reactive transport (Zhang et al., 2008). However, using these models, stiff PDEs develop when certain occurrences have very fast kinetics (close-to-equilibrium processes), making the direct approach unworkable. To overcome the challenges of the equilibrium reactions, a new method was introduced: the mixed differential and algebraic (DAE) approach (Yeh et al., 2005; Zhang et al., 2008). Yeh et al. (2005) first introduced this method, and before then no surface water quality model had completely embraced the mechanistic modeling of the transport of chemicals in rivers and streams, including both kinetic and very fast kinetic (equilibrium) reactions.

Studies on the water quality of the Philippine river systems mostly involve its physicochemical properties, e.g., Pleto, Migo & Arboleda (2020), Wiederhold, Tom Lichtenberg & Sarinas (2021), Escatron et al. (2022), Pasig River Coordinating and Management Office (PRCMO) (2023). These studies identify the values of water quality parameters and conclude whether the water quality is good or bad. However, it could not identify how the parameters are changing or how they are affected by different factors. Mathematical models involving these water quality parameters can provide an additional angle to the study of river water quality. This is important because river systems are dynamic systems. They change with respect to time, population, season, temperature, and more. Water quality models can help us look into the answers to the “how” and “why” of the water quality parameters. From there, we can conclude about the dynamics of a parameter, how it is changing, and why it is changing, and possibly we can look further into the future and estimate the values of the river water quality parameter. In the Philippines, the Pasig River was the birthplace of the old Manila civilization. The Pasig River served as an essential transportation route and water source for Spanish Manila. Because of neglect and industrial expansion, the river declined rapidly in the second part of the twentieth century and was pronounced biologically dead in 1990. Two decades after that announcement, a renaturation effort aimed to revitalize the river saw the restoration of life to the river (Villamor, 2009). It is now up to the current generation to restore the Pasig River to its former beauty.

Water quality models may be updated and improved to address new and developing surface water pollution issues, such as emissions from drainage and sewerage systems, due to the complex interactions brought about by growing human activity in the Pasig River (Rauch et al., 1998). With the use of the water quality model, we will be able to mathematically analyze the current state of the water quality in the Pasig River considering different factors affecting its conditions. The Yeh et al. (2005) model is best to use because it can handle a wide variety of river systems and at the same time work with small data sets. Moreover, we will be able to analyze the future status of the river and provide efficient resource management practices. Since the Pasig River is in the center of the NCR, the complexity of its water quality constituent (total dissolved solids, suspended sediments, chlorine, phosphate, etc.) is most likely applicable to the model being studied (Pasig River Coordinating and Management Office (PRCMO), 2023).

In this study, we have examined the model in Yeh et al. (2005) and Zhang et al. (2008) and incorporated the total dissolved solids (TDS) in mobile and immobile phases for the Pasig River. We do numerical simulations for the reactions of TDS in the Pasig River and how it changes with respect to time and position along the river. We also discuss the effect of the changing velocity, area, and perimeter of the Pasig River and different sources (sewerage system, tributaries) that directly drain into the river. With this, we employ sensitivity analysis to determine influential parameters on the TDS. The partial rank correlation coefficients (PRCC) technique is performed to determine how the TDS changes with respect to the changes in parameter values. We estimate the values of the influential parameters obtained during sensitivity analysis using the least squares method.

In the next sections, we discuss in summary the formulation of the considered water quality model from Yeh et al. (2005) and Zhang et al. (2008) and present the scheme to solve the model numerically and validate it. Lastly, the model is applied to the Pasig River data.

Considered model and methodology

In this section, we reiterate the model and the methodology from Yeh et al. (2005) and Zhang et al. (2008) which is the adopted model in this work. In the model, sediments are categorized as either suspended sediments (mobile) or bed sediments (immobile). Three forms of chemical species are considered in either the mobile or immobile phase, namely, dissolved chemicals, chemicals sorbed on sediment, and precipitates. Figure 1 illustrates the positions of chemical species and sediments in the cross-sectional area of a river.

Figure 1 The positions of chemical species and sediments in the cross-section of a river or stream (Zhang et al., 2008).

Following the development of the model from Yeh et al. (2005) and Zhang et al. (2008), the conservation law of material mass is used to develop the continuity equation for the constituents of mobile water quality. The law asserts that the rate of mass change is determined by both advective-dispersive transport and biogeochemical reactions, denoted as the reactive transport equation. In contrast, the balance equation for immobile water quality constituents is expressed as the rate of mass change solely influenced by biogeochemical reactions, denoted as the reaction equation. These equations are recast in the form

(1) ∂(AρiCi)∂t+αiL(ρiCi)=Ari,i∈M,

where ρi is the density of the phase, Ci is the concentration of water quality constituent i ( (Fm)i for mobile and (Fim)i for immobile), A is the cross-sectional area, ri is the reaction rate of water quality constituent i due to all biogeochemical reactions, and αi is 0 for immobile constituents and one for mobile constituents. We note here that the index set for all water quality constituents M is the disjoint union of the index sets for immobile water quality constituents and mobile water quality constituents.

The transport operator L (incorporating source terms) is given by

(2) L(ρiCi)=∂(QρiCi)∂x−∂∂x[AKx∂(ρiCi)∂x]−Si,

where the first term on the right-hand side is the advection term, the second term is the dispersion term, and Si are the sources/sinks of water quality constituent i. The sources could be a non-bank external source, rainfall source, overland source from river banks 1 and 2, or subsurface sources.

The reaction rate ri in a reaction-based formulation is determined by adding the rates of each individual reaction in the ith water quality constituent that takes part (Fang, Yeh & Burgos, 2003),

(3) ri=∑k=1nr[(νik−μik)rk],i∈M,

where nr is the total number of reactions, rk is the reaction rate of the kth reaction, μik is the reaction stoichiometry of the ith water quality constituent in the kth reaction associated with the reactants, and νik is the reaction stoichiometry of the ith water quality constituent in the kth reaction associated with the products. For simplicity, we have the equivalent expression for Eq. (1)

(4) U∂(AC)∂t+αL(C)=Avr,

where C=ρiCi, U is a unit/identity matrix, α is a diagonal matrix whose diagonal components are αi, v is the reaction stoichiometry matrix, and r is the reaction rate vector with the reaction rates as its components.

In the conventional approach, the distribution and variations of the constituents of water quality within its domain are obtained by directly solving Eq. (4). However, this approach is ineffective when an equilibrium reaction exists in the system. Thus, Yeh et al. (2005) investigated the decomposition technique for handling equilibrium reactions and the numerical method used to solve the system of equations that results from the decomposition. We refer the reader to Yeh et al. (2005) and Zhang et al. (2008) for the detailed discussion of the decomposition technique and the numerical methods that were used to solve the PDE.

The non-equilibrium equation obtained after decomposition together with the transport operator L becomes

(5) A∂Ek∂t+∂(QEkm)∂x−∂∂x[AKx∂Ekm∂x]=SEk+ARk,k∈NNE.

where Ek is the non-equilibrium variable, Ekm is the mobile part of the non-equilibrium variable, A is the cross-sectional area, Q is the flow rate, Kx is the dispersion coefficient, SEk’s are the different sources and sinks for each non-equilibrium variable Ek, and NNE is the index set of non-equilibrium variables. The equilibrium consistent equation together with Eq. (5) now form the system of algebraic and differential equations that will be solved to find the concentration of the water quality constituents. The basis of the study is this version of the model.

After decomposition, several methods are used to solve the resulting non-equilibrium variable Eq. (5). Generally, the finite element method (FEM) was the base method used and discussed in Yeh et al. (2005) in solving the derived system of PDEs in Eq. (5). FEM is one of the approaches used to compute approximate solutions to PDEs. It is a systematic method for estimating continuous functions using discrete models. This discretization involves dividing the space domain into finite subdomains, each of which forms a point known as a node. Finite elements are the non-overlapping subdomains that are connected at nodes on their boundaries. They carry piecewise and local approximations of the function, which are defined uniquely in terms of values held at their nodes (Tekkaya, Soyarslan & Reinhart, 2014). Meanwhile, a fully implicit scheme is a method that involves solving an equation involving the current state of the system as well as its next state in order to find numerical approximations to the solutions of time-dependent ordinary and partial differential equations.

The application of the fully implicit scheme on the time discretization and FEM for the space discretization of Eq. (5) over the domain of interest resulted in the following matrix equation

(6) ([L1]+[L2]+[L3]){(Ek)n+1}+[Z]{(Ek)n+1−(Ek)nΔt}={S0}+{B}

The matrices [L1], [L2], [L3], [Z] and the load vectors {S0} and {B} are given by

L1ij=−∫xixjdφidx[Q(EkmEk)n]φjdx,L2ij=∫xixjdφidx[KxA∂∂x((EkmEk)n)]φjdx,L3ij=∫xixjdφidx[KxA(EkmEk)n]dφjdxdx,Zij=∫xixjφiAφjdx,andS0i=∫xixjφi(SEk+ARk)dx,

where φi(x) is the linear basis function at each element. The load vector {B} describes the load vector for the boundary condition which means for the elements corresponding to the interior nodes, {B} is zero. To determine the boundary term {B} for the boundary node i=1,N, calculate as follows: Bi=Ek(xi,t) for the Dirichlet boundary condition and Bi=−nQEk(xi,t) for the variable boundary condition where n is the outward unit normal vector. At the upstream boundary node, the Dirichlet boundary condition will be applied, and at the downstream boundary node, the variable boundary condition. We note that when the flow comes in from outside nQ<0 and when the flow is going out from inside nQ>0. We also note that since two dependent variables are present in Eq. (5), namely Ek and Ekm, we express Ekm in terms of (Ekm/Ek)n⋅Ek to make Ek as the primary dependent variable. This explains the presence of (Ekm/Ek)n in the matrices L1, L2 and L3. Finally, Eq. (6) is used to find the individual water quality constituents.

Validation of the numerical solution

Before we apply the model to the Pasig River, we make sure to first validate the numerical solution. We consider a particular example in order to validate the presented numerical scheme in this section. In particular, a single chemical species is being modeled in two fluid phases, the mobile water phase with concentration Fm and the immobile water phase with concentration Fim. The concentrations Fm and Fim are considered to be related by the equilibrium reaction (R1):Fm⟺Fim,k=0.01. We assume that, initially, no chemical concentration exists in the domain of interest and no other sources except at the upstream boundary node. The methodology in the previous section was used for the model formulation of the species and the decomposition is also employed in the model. The resulting model is nondimensionalized and solved analytically using Laplace transform. The numerical solution discussed in the previous section was then used on the dimensionless model and the results were compared to the analytical solution. Python 3 was used to graph the results of the analytical and numerical solution of the nondimensionalized model.

We arrived at the following system of reactive transport equation and reaction equation

(7) ∂AρwFm∂t+L(ρwFm)=AR1

(8) ∂A⋅P⋅hb⋅ρwb⋅θb⋅Fim∂t=−P⋅hb⋅R1

Following the system formulation of Eqs. (1), (7) and (8) in matrix form become

(9) [1001][∂(AρwFm)∂t∂(AD1Fim)∂t]+[1000]L(ρw⋅Fm)=A[1−D2][R1],

where D1=P⋅hb⋅ρwb⋅θb/A, and D2=P⋅hb/A. Here the corresponding matrices U, α, and v compared with Eq. (4) are as follows:

U=[1001],α=[1000],v=[1−D2].

Since R1 is an equilibrium reaction, by Gauss-Jordan decomposition, Eq. (9) becomes

(10) [D2101][∂(AρwFm)∂t∂(AD1Fim)∂t]+[D2000]L(ρw⋅Fm)=A[0−D2][R1].

From Eq. (10), we get

(11) ∂(A⋅ρw⋅P⋅hb⋅Fm)∂t+∂(A⋅P⋅hb⋅ρwb⋅θb⋅Fim)∂t+L(ρw⋅P⋅hb⋅Fm)=0

(12) ∂(P⋅hb⋅ρwb⋅θb⋅Fim)∂t=−P⋅hb⋅R1.

Furthermore, because R1 is an equilibrium reaction and Eq. (12) contains R1, Eq. (12) is replaced by the thermodynamic equilibrium Eq. (13)

(13) ∂(P⋅hb⋅ρwb⋅θb⋅Fim)∂t=R1≈∞⇒κ=FimFm⇒Fim=0.01Fm.

Equation (11) is now considered as reactive transport for non-equilibrium variables equation. For convenience, we still denote Fm=ρw⋅P⋅hb⋅Fm and Fim=P⋅hb⋅ρwb⋅θb⋅Fim and explicitly define the transport operator L in Eq. (2) into Eq. (11). We then have

(14) ∂(A⋅Fm)∂t+∂(A⋅Fim)∂t+∂(QFm)∂x−∂∂x[AKx∂(Fm)∂x]=0.

Equation (14) is the final equation that we are going to solve using the finite element method and the fully implicit scheme together with the corresponding thermodynamic equilibrium equation.

Analytical solution to Eq. (14)

To define an analytical solution to Eq. (14) we first do nondimensionalization. Often, differential equations that show up in modeling real-world phenomena contain many constants. These constants have different units, which can complicate the analysis. Nondimensionalization is the first and arguably the most important step in the analysis of a system of differential equations. It involves choosing appropriate units for the variables in the problem to reduce the number of constants leaving a dimensionless variable. By introducing dimensionless variables, we can derive a simplified differential equation and later determine the most convenient choice of units. Nondimensionalization offers a number of advantages. One of which is that the solution of a nondimensional PDE is universal and encompasses an infinite number of solutions whereas the dimensional solution is limited to a single set of parameter values (Goltz & Huang, 2017). The nondimensional model is now solved using Laplace transform. Laplace transform is a widely used technique for solving differential equations. Laplace transform is an integral transform that converts a function of a real variable to a function of a complex variable. The popularity of the Laplace transform lies in how easy it is to implement because of the reduction of a differential equation into an algebraic problem.

The dimensional dependent and independent variables in Eq. (14) are now replaced with dimensionless variables that we specify. We divide the dimensional variable by an arbitrary constant of the same dimension to define these dimensionless variables in Eq. (14). For example, for Fim, we choose an arbitrary constant ν0 with the same unit as Fim so that the variable ν=Fim/ν0 is dimensionless. In the same manner, we can choose arbitrary constants μ0, ξ0 and τ0 so that the variables μ,ξ and τ are dimensionless where μ=Fm/μ0, ξ=x/ξ0 and τ=t/τ0. Then

(15) Aμ0τ0∂(μ)∂τ+Aν0τ0∂(ν)∂τ+Qμ0ξ0∂(μ)∂ξ−KxAμ0ξ02∂∂ξ(∂μ∂ξ)=0.

Divide by the constant of the fourth term in Eq. (15), we have

(16) ξ02τ0Kx∂(μ)∂τ+ν0ξ02τ0Kxμ0∂(ν)∂τ+Qξ0KxA∂(μ)∂ξ−∂∂ξ(∂μ∂ξ)=0.

Since ξ0 and τ0 are arbitrary constant, we can set ξ0=KxAQ and τ0=KxA2Q to simplify Eq. (16) into

(17) ∂∂τ(μ+ν0μ0ν)+∂(μ)∂ξ−∂∂ξ(∂μ∂ξ)=0.

Finally, to follow the notation of the previous section we let η=μ+ν0μ0ν be the non-equilibrium variable and ηm=μ be the mobile part of the non-equilibrium variable, then Eq. (17) becomes Eq. (18)

(18) ∂η∂τ+∂ηm∂ξ−∂2ηm∂ξ2=0.

The Dirichlet boundary condition is ηm(0,τ)=η0m. For the initial condition, we assume that there is no chemical present at τ=0, i.e., ηm(ξ,0)=0 and η(ξ,0)=0. Following the methodology described in the previous section, make η become the primary dependent variable by letting ηm=(ηmη)nη. Now Eq. (18) will become

(19) ∂η∂τ+γ∂η∂ξ−γ∂2η∂ξ2=0,

where γ=(ηmη)n is calculated from the concentration at the previous time step.

Using the Laplace transform defined in Goltz & Huang (2017), we convert the PDE Eq. (19) with its initial and boundary condition into an ordinary differential equation (ODE) with boundary condition in “Laplace time”. We then will invert the Laplace time solution back into “real-time” to obtain the solution for concentration as a function of ξ and τ. To begin, we define the transform function η¯(ξ,s) where s is the Laplace variable as η¯(ξ,s):=L{η(ξ,τ)}=∫0∞η(ξ,τ)e−sτdτ. We apply the Laplace transform to Eq. (19) together with its initial and boundary condition and obtain the following ODE

(20) sη¯(ξ,s)+γdη¯(ξ,s)dξ−γd2η¯(ξ,s)dξ2=0.

For the boundary condition L{η(0,τ)}=η¯(0,s)=η0ms, where η0m is the original Dirichlet boundary condition. For more details on the method of Laplace transform defined above, see Appendix A of Goltz & Huang (2017). Hence, the general solution of Eq. (20) is

(21) η¯(ξ,s)=k1η¯1(ξ,s)+k2η¯2(ξ,s),

for some constants k1 and k2, where

(22) η¯1(ξ,s)=exp((12+γ2+4γs2γ)ξ)and

(23) η¯2(ξ,s)=exp((12−γ2+4γs2γ)ξ).

Since the exponent of η¯1(ξ,s) in Eq. (22) is positive, its derivative also has a positive exponent, hence, as ξ→∞ the term would be infinite. Thus, k1 must be zero. To obtain a particular solution, apply the boundary condition into Eq. (21), that is, when ξ=0, we have η¯(0,s)=k2η¯2(0,s) and so η0ms=k2. Thus, the particular solution to Eq. (20) is given by

(24) η¯(ξ,s)=η0mse(12−γ2+4γs2γ)ξ.

The solution Eq. (24) is in Laplace time. To obtain the solution in real-time, we must invert it. The simplest way to invert the Laplace time solution Eq. (24) is to use Laplace inversion tables. See appendix E of Goltz & Huang (2017) for the useful Laplace transform. The term on the right-hand side of Eq. (24) can be rewritten as follows

(25) η0mse(12−γ2+4γs2γ)ξ=η0me12ξe−1γξγ4+s(s+γ4)−γ4,

which is similar to one of the forms in the Laplace inversion table. Note that another property of Laplace transforms is that the Laplace inversion of η¯(ξ,s+a) is e−atνn. Using the Laplace inversion table, the inverse of Eq. (25) is

(26) η0me12ξ⋅e−γ4τ⋅12[e(γ4τ−γ4⋅1γξ)erfc(1γξ2τ−γ4τ)+e(γ4τ+γ4⋅1γξ)erfc(1γξ2τ+γ4τ)]=η0m2[erfc(ξ2γτ−γ4τ)+eξerfc(ξ2γτ+γ4τ)].

The erfc(x) is the complementary error function of x defined as erfc(x)=2π∫x∞e−t2dt. Finally, the fully inverted expression for the Eq. (24) is

(27) η(ξ,τ)=η0m2[erfc(ξ2γτ−γ4τ)+eξerfc(ξ2γτ+γ4τ)].

Equation (27) is then visualized using a Python 3 program. Note that in Eq. (17) we have the following: ηm=μ, ν=0.01μ0ν0μ, η=1.01μ, and γ=0.99. Together with the boundary condition η0m=1, the graph of Eq. (27) is provided in Fig. 2.

Figure 2 Comparison of the analytical and numerical solutions of Eq. (19) at τ=25,50.

Numerical solution of Eq. (19)

Following the procedure discussed above, by the fully implicit scheme, Eq. (19) we have

(28) η−ηnΔτ+γ∂η∂ξ−γ∂2η∂ξ2=0,

where the terms without superscripts correspond to time step n+1. We apply FEM for the spatial discretization of Eq. (19) (Fletcher, 1984) over the domain [x1,xN] where x1 and xN represents the first and last node, respectively, at an arbitrary time step t we can approximate the solution η by a linear combination of the basis functions ηt≈∑j=1Nηjtφj(x), where ηjt are the nodal values of η at the jth node and time t and φj(x) is the linear basis functions at each element. Thus, the following matrix equation is obtained

(29) [Z]{η−ηnΔτ}+(γ[A]+γ[B]){η}=0

where Zij=∫φiφjdξ, Aij=−∫dφidξφjdξ, and Bij=∫dφidξdφjdξdξ. Following the procedure above we obtained the graph of the solution Eq. (29) shown in Fig. 2 with Dirichlet boundary condition η0m=1, element size is 1 unit and the time step size is 1 unit. The total space domain is 100 and the total simulation time is 50. Finally, Fig. 2 shows the comparison of the analytical and numerical solutions of the Eq. (19) at time τ=25,50. Moreover, Fig. 3 shows the absolute errors between obtained solutions in Fig. 2. The computed errors are relatively small and therefore, we can say that we determined a good approximation.

Figure 3 The computed absolute error between the analytical and numerical solution at τ=25,50.

Application to the pasig river

In this section, we show the applicability of the model to the Pasig River, that is, the presentation of the data, the obtained reactive transport equation of the non-equilibrium variable, and our assumptions for the model. This section also presents the results and discussion of the simulations using the Pasig River data, which include sensitivity analysis and parameter estimation to further improve the results of the model. All the code for the simulations in this section is available in Abas (2024).

Pasig River system and PRUMS data

An interagency project called the Pasig River Unified Monitoring Stations (PRUMS) program aims to standardize data and monitoring stations for water quality in the Pasig River System and provide logical reports for the information of the general public. It is under the direct supervision of the DENR and the PRCMO (Pasig River Coordinating and Management Office (PRCMO), 2023).

In our study, we consider three stations along the Pasig River-Bambang station, Guadalupe station, and Lambingan station (see Figs. 4 and 5). Stations like Vargas station, Guadalupe Nuevo, Buayang Bato, Guadalupe Viejo, and Havana station are treated as sources since these are tributaries that flow directly into the main Pasig River. We chose only Bambang to Lambingan Stations because the data for the tributaries (sources) along these stations are available. On the other hand, the station before Bambang is too close to Laguna Lake and the stations beyond Lambingan are too close to Manila Bay and there exist many tributaries that are not considered in our simulations due to the unavailability of the data. Tributaries, sea tides, and other sources might affect the concentration of a water quality constituent yet cannot be included in our simulation. Bambang station and Lambingan station are approximately 8,500 meters (m) away from each other and Guadalupe Station is about 3,600 m down from Bambang station.

Figure 4 The map for the Pasig River Unified Monitoring System (PRUMS) stations taken from Pasig River Coordinating and Management Office (PRCMO) (2023).

Maps Data: ©2024 Google/This map includes data from: Airbus Data SIO, NOAA, U.S. Navy, NGA, GEBCO (Google, 2023).

Figure 5 PRUMS stations of interest along the Pasig River.

Since there is no available data for the parameters in between stations, we take their values as the average value of the parameters of the two adjacent stations. Moreover, for transient simulation, Bambang Station is considered a Dirichlet boundary condition. Hence, our results and discussion focus more on the two stations—Guadalupe and Lambingan.

This study focuses on the concentration of total dissolved solid (TDS) in the Pasig River, which occurs in two fluid phases; the mobile phase with concentration Fm and the immobile phase with concentration Fim. We assume that their relationship is the same as stated in the previous section, that is, Fm⟺Fim,k=0.01. TDS is the dissolved total content of all organic and inorganic materials in a liquid that is suspended in molecular, ionized, or microgranular form. The main components are often carbonate, hydrogen carbonate, chloride, sulfate, and nitrate anions together with calcium, magnesium, sodium, and potassium cations (World Health Organization, 1996). The space domain of the study is only from Bambang Station to Lambingan Station, which is approximately 8,500 m, as shown in Fig. 5. The density, specific gravity, and porosity are assumed to be constant while the flow velocity, boundary condition, perimeter, and depth change every month.

The reactive transport of the non-equilibrium variable of the TDS of the Pasig River has been derived following the procedure in the methodology section and is given by

(30) ∂(A⋅Fm)∂t+∂(A⋅Fim)∂t+∂(Q⋅Fm)∂x−∂∂x(Kx⋅A⋅∂Fm∂x)=0.

Again, for simplicity, we set Fm=ρw⋅P⋅hb⋅Fm, Fim=P⋅hb⋅ρwb⋅θb⋅Fim, A as the cross-sectional area, Q as the flow rate of the water, and Kx as the dispersion coefficient. Equation (30) is accompanied by the thermodynamic equilibrium equation Fm=0.01Fim.

The rationale for choosing TDS as the considered chemical species is its being one of the important chemical parameters in water quality monitoring. The process of measuring the total dissolved solids (TDS) in freshwater involves filtering the water using a 2 micrometer ( μm) filter, letting the filtrate evaporate until it reaches dryness, and then reporting the weight of the solids that remain in grams (g) per liter (L) (Boyd, 2020). In most definitions, the total dissolved solids limit is 0.5−1.0 g/L. Thus, some inland waters are saline waters rather than fresh waters. Drinking water should not exceed about 0.5 g/L of total dissolved solids (Boyd, 2020). The diffusion coefficient Kx is unavailable in the data so we set Kx=1,000. The parameters porosity and density are also assumed and are equal to 1, the area and wetted perimeter can be computed from the available data (Pasig River Coordinating and Management Office (PRCMO), 2023), while the distance between stations of the Pasig River and the width of the river are measured from Google Maps (Google, 2023). Table 1 summarizes the assumed parameter values in this study.

Table 1 Description and assumed values of the parameters for the Pasig River simulation.

Parameter	Description	Assumed values	
ρw	Density of water column (g/L)	0.01	
ρwb	Density of water bed (g/L)	0.01	
θb	Porosity	1	
Kx	Dispersion coefficient (m2/s)	1,000	
hb	Riverbed depth (m)	1	
w	Width of the river (m)	70–85	

We use the fully implicit scheme and the finite element method to solve the advective-dispersive equation with a fixed time step size of 360 seconds (s) and finite element mesh size of 50 meters (m).

Preliminary results

As stated above, the Bambang station is considered a boundary station, so the actual data observed at the Bambang station is the initial concentration with the Dirichlet boundary condition (Zhang et al., 2007). With the assumed value for Kx and some other parameters, we can see in Figs. 6 and 7 the difference between the observed data and the simulated data at the two stations. In this study, 3 years of data have been observed that cover the years 2018, 2019, and 2021. Not much data was available for the year 2020 because of the COVID-19 pandemic. The Pasig River data for flow velocity and TDS concentration can be found in the Tables A1–A4.

Figure 6 Comparison of the approximated concentration of TDS using the model at the Guadalupe station to the actual data.

Figure 7 (A–C) Comparison of the approximated concentration of TDS using the model at the Lambingan station to the actual data.

Table A1 Concentration of TDS in grams per liter (g/L) at the three considered stations in the Pasig River (NA–not available).

Month	TDS concentration (g/L)	
	Bambang	Guadalupe	Lambingan	
	2018	2019	2021	2018	2019	2021	2018	2019	2021	
January	0.27	0.27	0.539	0.27	0.28	0.479	0.25	0.28	0.444	
February	0.33	0.27	0.42	0.32	0.28	0.48	0.30	0.32	0.39	
March	0.32	0.31	0.49	0.31	0.32	0.47	0.35	0.30	0.46	
April	0.37	0.50	0.46	0.37	0.70	0.43	0.87	1.95	0.36	
May	9.13	6.54	0.40	16.28	9.48	0.49	22.25	12.06	0.55	
June	0.53	4.33	0.45	0.48	9.16	0.48	0.48	14.76	0.51	
July	0.32	0.46	0.45	0.31	0.56	0.42	0.32	6.29	0.48	
August	0.34	0.49	0.37	0.33	0.35	0.32	0.32	0.36	0.31	
September	0.32	1.74	0.30	0.32	1.50	0.33	0.32	1.60	0.31	
October	0.26	0.46	1.36	0.29	0.46	1.07	0.28	0.46	0.84	
November	0.26	0.26	0.30	0.26	0.48	0.29	0.28	0.58	0.32	
December	0.27	NA	0.30	0.27	NA	0.76	0.28	NA	0.30	
Note:

Bold values are missing values from the PRUMS report, these values are either taken from the same month of the previous year or are taken from the preceding month.

Table A2 Concentration of TDS in grams per liter (g/L) in the tributaries along Pasig River (NA–not available).

Month	TDS concentration (g/L)	
	Vargas	Buayang Bato	Guadalupe Nuevo	
	2018	2019	2021	2018	2019	2021	2018	2019	2021	
January	0.30	0.34	NA	0.44	0.36	NA	0.44	0.48	0.297	
February	0.41	0.42	NA	0.44	0.38	NA	0.51	0.45	0.46	
March	0.43	0.51	NA	0.41	0.48	NA	0.48	0.57	0.46	
April	0.43	0.98	NA	0.40	0.37	NA	0.52	0.51	0.44	
May	4.25	4.45	NA	2.98	0.35	NA	15.00	0.50	0.51	
June	0.30	3.58	NA	0.47	4.98	NA	0.57	0.92	0.45	
July	0.21	0.37	NA	0.37	0.25	NA	0.47	0.61	0.51	
August	0.19	0.21	NA	0.40	0.34	NA	0.45	0.48	0.34	
September	0.18	0.00	NA	0.37	0.00	NA	0.35	0.00	0.48	
October	0.22	0.30	NA	0.34	0.36	NA	0.42	0.49	0.51	
November	0.30	0.01	NA	0.30	0.41	NA	0.41	0.50	0.45	
December	0.27	NA	NA	0.33	NA	NA	0.46	NA	0.99	
Note:

Bold values are missing values from the PRUMS report, these values are either taken from the same month of the previous year or are taken from the preceding month.

Table A3 Concentration of TDS in grams per liter (g/L) in the tributaries along Pasig River (NA–not available).

Month	TDS concentration (g/L)	
	Guadalupe Viejo	Havana	
	2018	2019	2021	2018	2019	2021	
January	0.41	0.47	0.352	0.38	0.47	0.386	
February	0.49	0.50	0.41	0.47	0.56	0.39	
March	0.48	0.55	0.38	0.57	0.49	0.40	
April	0.51	0.51	0.43	0.64	0.57	0.37	
May	18.08	0.80	0.42	0.55	0.52	0.42	
June	0.54	2.00	0.43	0.54	0.51	0.34	
July	0.42	0.44	0.42	0.47	0.37	0.36	
August	0.37	0.43	0.32	0.38	0.48	0.18	
September	0.39	0.00	0.35	0.45	1.61	0.33	
October	0.34	0.46	0.62	0.40	0.46	0.86	
November	NA	0.54	0.36	NA	0.68	0.38	
December	0.43	NA	0.36	0.45	NA	0.32	
Note:

Bold values are missing values from the PRUMS report, these values are either taken from the same month of the previous year or are taken from the preceding month.

Table A4 Flow velocity in meters per second (m/s) at the three considered stations along the Pasig River (NA–not available).

Month	Flow velocity (m/s)	
	Bambang	Guadalupe	Lambingan	
	2018	2019	2021	2018	2019	2021	2018	2019	2021	
January	0.59	0.11	1.22	0.07	0.02	0.87	0.53	0.62	0.1	
February	0.27	0.91	0.14	0.43	0.06	0.53	0.02	0.45	2.18	
March	0.47	0.46	1.21	0.56	0.12	0.11	0.43	0.08	0.03	
April	0.23	0.01	0.13	0.11	0.04	0.12	0.32	0.32	0.12	
May	0.64	0.36	0.35	0.3	0.03	1.04	0.26	0.05	1.15	
June	0.36	0.54	0.94	0.22	0.04	1.09	0.30	0.50	0.12	
July	1.03	0.59	0.14	0.70	0.03	0.29	0.84	0.03	0.28	
August	0.24	0.08	0.22	0.77	0.60	0.15	0.75	0.51	0.24	
September	0.36	0.73	0.31	0.81	0.12	0.28	0.82	0.65	0.26	
October	1.25	1.05	0.31	0.28	0.48	0.24	0.79	0.66	1.40	
November	1.08	0.88	0.12	0.01	0.20	0.95	0.64	0.61	0.12	
December	0.67	NA	0.23	0.30	NA	0.87	0.53	NA	0.14	
Note:

Bold values are missing values from the PRUMS report, these values are either taken from the same month of the previous year or are taken from the preceding month.

The data shows that in 2018 and 2019, the TDS concentration significantly increased in April, May, and June (see Figs. 6 and 7). The observation holds for both the Guadalupe and Lambingan stations. Furthermore, the months that show an increase in concentration are considered to be the dry season of the year. During the dry season, there is a decrease in water level due to the extreme heat and this may have contributed to the concentration increase of TDS. This means that if the same concentration is introduced into a system with a smaller volume, the concentration will become more dense and an increase in concentration will be observed. Nevertheless, our observation does not hold for the same months in the data for 2021. The fact that the TDS concentration did not rise throughout the 2021 dry season suggests that the drop in water level had no bearing on the concentration rise. This indicates that in 2021, TDS levels were lower. This is understandable given that 2021 was still a pandemic year in the Philippines and that very few people were permitted to leave the house. The Pasig River may have had lower TDS levels as a result of less movement and activity in the area.

Other factors could have caused the data spike. It is likely that certain activities (such as dredging, clean-up drives, abrupt rises in water, rain, and more) took place in the area before or on the day of the sampling, upsetting the water bed and raising the TDS concentration. This is less likely to be the case, though, given that the data with higher values are consistent with the two stations that were recorded in 2018 and 2019. It would have made sense for the TDS concentration to rise during the dry season in 2021, but it did not rise because of the COVID-19 pandemic restrictions. Apart from the potential reason for the increase, it is also plausible that there exist unidentified sources of TDS concentration that were overlooked in the model or during the simulation. This is significant since it will tell researchers whether or not the unknown source has a significant impact on the TDS concentration.

The simulation results exhibit the appropriateness of the model. Even when the actual data increases or decreases, the simulated data follows accordingly. This implies that the model considered even extreme data values. Still, there are months in which there is a substantial difference between the simulated and real data. In the model, the flow velocity or flow rate has a significant impact on the model output, which is the TDS concentration; see the sensitivity analysis section. In the station of interest, there are two flow rates to consider: inflow and outflow. Naturally, the inflow increases the TDS content in the stations, while the outflow reduces it. If the inflow is fast and the flow out is slow, the TDS concentration rises because it enters the system quickly and exits slowly. On the other hand, the TDS concentration falls because it enters the system slowly and exits rapidly. This is how the model output behaves during simulation in both stations. Figure 6B in May (2018), Fig. 6C in March (2021), Fig. 7A in May (2018), Fig. 7B in May, June, and July (2019), and Fig. 7C in January (2021) are examples of cases where the flow rate and TDS values do not reflect this behavior, see Fig. 8 for the comparison of the computed flow rates between stations. These are the months/cases where the actual data and simulated data differ significantly. Table 2 shows the relative errors between the actual and simulated values of the TDS in both Guadalupe and Lambingan Station for each month. It can be observed that the errors are quite big in this simulation. This might be due to the assumed values of some parameters, such as the flow rate and dispersion coefficient.

Figure 8 Computed flow rates in the three stations along the Pasig River.

Table 2 Relative errors between the simulated and actual data for each month at Lambingan and Guadalupe stations (NA–not available).

Month	Relative error	
	Guadalupe	Lambingan	
	2018	2019	2021	2018	2019	2021	
January	0.203	0.175	1.027	0.385	0.793	4.768	
February	1.146	1.236	0.465	1.242	0.108	0.665	
March	0.363	0.642	8.442	0.217	0.056	2.166	
April	0.322	0.281	0.353	0.884	0.926	0.645	
May	0.089	1.317	0.649	0.559	0.647	0.760	
June	0.758	0.219	0.037	0.286	0.878	1.835	
July	0.917	1.575	0.293	0.209	0.908	0.607	
August	0.182	0.449	0.082	0.415	0.630	0.695	
September	0.304	0.157	0.216	0.483	0.644	0.675	
October	3.579	0.862	0.659	0.791	0.379	0.758	
November	0.741	0.249	0.277	0.513	0.744	0.195	
December	0.330	NA	0.520	0.715	NA	0.078	

As researchers and users of secondary data, we consistently assume that the measurement tools are correctly calibrated and that the data are real and accurate. Therefore, in situations where there is a substantial difference between the simulated and real data, there may be other parameters or factors influencing the real data during the sampling time that were overlooked during model formulation. Note that the model formulation includes the following parameters: area, dispersion coefficient, river depth, riverbed depth, width, flow velocity, density, and porosity. In addition, we considered many sources of TDS when developing the model. Factors influencing the increase or decrease in actual TDS values in the aforementioned months/cases in the previous paragraph are no longer within the parameters and sources under consideration. Rain is an important aspect since it adds a lot of water to the river. Dredging efforts may also be a factor since they alter riverbed sediments, the sewerage system, and other human activities along the river. We are unable to establish the effect of these factors on our real and simulated data due to the lack of available data. In addition, the different sources (tributaries going straight into the main Pasig River) along the Pasig River have little to no effect on the concentration of TDS in all stations, most significantly at Guadalupe Station where there are three sources of inflow (see Fig. 5). This is because these tributaries have low water discharge and TDS concentrations.

The cases mentioned above, in which there is a substantial difference between the simulated and real data, should be emphasized because this is the point at which the model results diverge from the real data. Comparing Figs. 6 and 7 closely reveals that the majority of the plots representing the simulated data closely resemble the real data. This suggests that our model formulation of the river system is appropriate. In these cases, discrepancies between our results and the real data could be attributed to the assumptions made during the numerical implementation of the model, including those regarding the dispersion coefficient Kx, density, porosity, and the values of the parameters between stations, such as the dispersion coefficient, flow rate, depth, and width. These assumptions were made because the data are only available in the station of interest but not throughout the river system. With these observations, the next section discusses the identification of key parameters that greatly affect the concentration of TDS along the Pasig River.

Sensitivity analysis

Sensitivity analysis (SA) gives users of simulation and mathematical models the ability to evaluate the level of model adequacy and identify which parameters have the most or least impact on the output of the model. As a result, critical parameters on the model output must be given precise values, whilst less critical parameters only require a rough approximation (de los Reyes & Escaner, 2018). The goal of SA is to determine the important input of the model (parameters and initial conditions) and quantify how input uncertainty affects model outcome(s). This sensitivity measure is simply computed numerically by executing many simulations adjusting input components around a nominal value (Marino et al., 2008). The partial rank correlation coefficient (PRCC), which has been shown to be the most dependable and effective sampling-based technique, was employed as the sensitivity analysis method in this work (de los Reyes & Escaner, 2018).

Latin hypercube sampling and partial rank correlation coefficient

The methodology and numerical implementation of this method was based on the SA on biomathematical ODE models of Blower and Dowlatabadi (Marino et al., 2008). Partial rank correlation coefficient (PRCC) values are determined for each input variable and each outcome variable in the following manner.

Latin hypercube sampling (LHS) performs sampling independently for each parameter p. The uniform parameter distribution is divided into H equal probability intervals. Sampling is done by randomly selecting values from each interval. Each interval for every parameter is sampled exactly once (without replacement), allowing the whole range for each parameter to be investigated (Marino et al., 2008). As a result, the (H×p) LHS matrix is created, with H rows representing the number of simulations (sample size) and p columns representing the number of varied parameters.

Define the model output function as y=f(x,t,θ) where y is the model outcome, x is the space domain, t is the time domain, and θ∈Rp is the parameter variable. Next, generate the outcome vector yi=f(x0,t,θi) where x0 is the specific position in the space domain and θi is a row input vector from the LHS matrix. Then, the LHS matrix, which is now H×(p+1) in size, receives an additional column containing the outcome vector yi. In each of these columns, the ordinal numbers corresponding to the rank ( 1 to H) are defined as the set (ri1,ri2,…,rik,Ri). It is now possible to define a p+1 by p+1 correlation coefficient symmetric matrix C, whose entries are cij.

cij=∑tH(rit−μ)(rjt−μ)∑tH(rit−μ)2∑sH(rjs−μ)2,i,j=1,2,…,p,

where μ=(1+H)/2. For the ci,p+1 elements, Ri replaces rjt and rjs. The leading diagonal elements of C are all ones. Define the matrix B as the inverse of C.

B=[bij]=C−1.

The PRCC (γiy) between the ith input parameter and the yth outcome variable is defined as Kendall & Stuart (1979)

γiy=−bi,p+1biibp+1,p+1.

Implementation in Python

Recall that the continuity Eq. (30) for the non-equilibrium variable equation can be written as

(31) ∂(A⋅Fm)∂t+∂(A⋅Fim)∂t+∂(Q⋅Fm)∂x−∂∂x(Kx⋅A⋅∂Fm∂x)=0.

To simplify the process, we consider Fm as the model output. Hence, there are only three parameters involved, the area, dispersion coefficient Kx, and the flow rate Q. The flow rate Q along the river has been divided into three regions: Q1 in Bambang, Q2 in Guadalupe, and Q3 in Lambingan Station. Parameters are then described in Table 3. The area is calculated based on the data from Pasig River Coordinating and Management Office (PRCMO) (2023) and will not be included in the sensitivity analysis since it will be part of the flow rate Q. The range of the dispersion coefficient is based on the values used by Zhang et al. (2008) in their article. The flow rates Qi are fitted using the least square method (see parameter estimation section).

Table 3 Description and values of the parameters

Parameter	Description	Range	Reference	
A	Cross-sectional area (m2)	[0, 300]	Calculated (Pasig River Coordinating and Management Office (PRCMO), 2023)	
Kx	Dispersion coefficient (m2/s)	[0, 1,000]	Assumed (Zhang et al., 2008)	
Q1	Flow rate at Bambang Station (m3/s)	[0, 400]	Data-fitted	
Q2	Flow rate at Guadalupe Station (m3/s)	[0, 400]	Data-fitted	
Q3	Flow rate at Lambingan Station (m3/s)	[0, 400]	Data-fitted	

The PRCC values range from −1 to 1. Positive (negative) values show a positive (negative) correlation between the parameter and the model output. It is implied by a positive (negative) correlation that a positive (negative) change in the parameter will cause the output of the model to rise or fall. The correlation of the parameter with the output increases with the absolute value of the PRCC. To obtain PRCC values, LHS is chosen for the input parameters. The area A is not included since the area is part of the parameter Q. The range for each parameter is shown in Table 3. The model output is the TDS concentration Fm. The number of simulations performed is 500, wherein a set of parameter values are selected in each simulation from a uniformly distributed values of the parameter range. Node x=72 and x=170 in the discretization of the model are the space points of interest. Time points of interest are chosen in order to investigate the effects of parameter changes on the output of the model. Figure 9 shows the PRCC values of the model output at x=72 and t=100 in 2018.

Figure 9 Guadalupe station: PRCC of model parameters at x=72 and t=100 with TDS concentration as the model output.

In Fig. 9, the parameter Kx has values closer to zero, indicating that the dispersion coefficient is less sensitive to the model output. This result is reinforced by Zhang et al. (2008), where the simulation is advection-dominated, with dispersion having a lesser effect on the function value. This demonstrates that this simulation is dominated by the flow of the water rather than the spread of the TDS. That is, the parameter flow rate Q is more sensitive to the model output; more specifically, Q1 has a positive correlation, but Q2 and Q3 have negative correlations. Given that the point of interest is at x=72, which is in Guadalupe Station, the results of the SA analysis make sense. While Q2 and Q3 flow away from the point of interest and will have a negative effect on the model output, Q1, which is in Bambang Station, flows directly toward the point of interest and has a positive impact on the model output. Changing the point of interest in the model output to x=170, which is at Lambingan station, Q2 will now have a positive impact on the model output, hence, PRCC values are positive (see Fig. 10).

Figure 10 Lambingan station: PRCC of model parameters at x=170 and t=100 with TDS concentration as the model output.

To see how the PRCC values change over time, the PRCC values are calculated for multiple time points and plotted vs. time. Figure 11 allows us to assess the significance of the parameters over the entire time interval for January. Since the parameters are constant throughout the month, they will greatly affect the model output mostly during the start of the simulation but eventually, after some time the values stabilized and assumed near constant PRCC values. This makes sense because the initial condition states that no TDS is present at the start of the simulation. Thus, all of the parameters will affect the TDS at the start. Figure 12 is a zoomed-in graph of Fig. 11, which clearly shows that the PRCC values started stabilizing at t=150. Lastly, the other months of 2018 also show the same behavior as the January results. PRCC results in a specific time step are enough to determine the sensitivity of a parameter.

Figure 11 PRCC of model parameters at x=72 calculated for multiple time points and plotted versus time (January) with TDS concentration as the model output.

Figure 12 PRCC of model parameters at x=72 for calculated for the time points interval [0,300] and plotted vs. time (January) with TDS concentration as the model output.

The sensitivity analysis concludes by showing that the parameter flow rate gained higher PRCC values in all stations suggesting a significant influence on the TDS concentration of the Pasig River. Conversely, the dispersion coefficient exhibits smaller PRCC values, indicating a reduced impact on the concentration of TDS in the Pasig River. In view of these results, it is necessary to determine the flow rate accurately, whereas an appropriate estimate of the dispersion coefficient suffices (de los Reyes & Escaner, 2018). To obtain precise values of the critical parameters, parameter estimation is applied. This is covered in detail in the section that follows.

Parameter estimation

The flow rate Q of the river can be found by obtaining the product of the flow velocity and the cross-sectional area of the river. Although the values of the flow velocities are available, there might be inconsistencies in the cross-sectional area of the river. Here we just assume that the shape cross-sectional area of the Pasig River is rectangular with a width w and depth h. However, this might not be true throughout the river in the domain of interest since there is a possibility for varied w or riverbeds that are not horizontally level. Moreover, since flow velocity is not available between stations, the observed flow velocity in the three stations might be affected by some factors, such as blockage of the flow due to garbage pollution, large objects, or the presence of water lilies, that hinder the flow. The flow can also be affected by the tributaries and/or unknown sewerage systems. Hence, there is a disruption or discontinuity in the flow velocity or flow rate.

Thus, we are interested in estimating the flow rate. The sensitivity analysis shows that the flow rate Q greatly affects the model output (TDS) more than the dispersion coefficient Kx. Specifically, the dispersion coefficient has a PRCC value that is close to zero, indicating that K has less effect on the model output which is the TDS, see Fig. 9. Our goal is to find values for the flow rate Qij per month i in each station j, j= Bambang, Guadalupe, and Lambingan, for which the error between observed concentration and the simulated concentration of TDS is minimized. The objective function is

(32) Oi(Qij)=||Fi(Qij)−Fijobs||22

where Fijobs is the observed concentration each month i in three stations j, concentrations Fi(Qij) are computed each month i in three stations by the PDE in Eq. (30), and ||⋅||2 is the Euclidean norm. The least squares method is used to minimize the objective function Eq. (32) for some initial values Q0j=100 of Qij for all j. The dispersion coefficient Kx is set as Kx=1,000.

Figure 13 shows the results of the parameter estimation for Qij and it presents two interesting outcomes. The first is a negative flow rate at Lambingan Station in 2019, while the second is an immense flow rate in Guadalupe station in all 3 years compared to the computed flow rate.

Figure 13 Comparison of the estimated (using parameter estimation: A, C, E) and computed (using the formula: B, D, F) flow rates in the three stations (Bambang, Guadalupe, Lambingan) for 3 years (2018: A&B, 2019: C&D, 2021: E&F).

To reduce the error between the actual and simulated values, the least squares program generates a negative flow rate. The negative flow rate simply indicates that the flow switches direction from the normal direction of the flow of the river. This makes sense for Lambingan Station, which is located close to Manila Bay and might be affected by the tide. During this period, the tide may be high enough to reach Lambingan Station. Although it had no effect on the flow rate during the sampling day, it may have influenced the TDS concentration, causing the least squares program to report a negative flow rate.

In the case with an immense flow rate, comparing the computed rates to the estimated flow rates, the Guadalupe Station shows the biggest differences. In the report of the Pasig River Unified Monitoring Stations, the actual data is the flow velocity of the river water and not the flow rate (discharge). However, the cross-sectional area is not provided and so the computed flow rate in Figs. 13B, 13D, 13F are subject to the errors of the computations in the cross-sectional area. Although measurements for the river depth have been provided, we are assuming that the depth is equal throughout the river width, resulting in a rectangular cross-sectional area while the river width is only based on the measurement in Google Maps (Google, 2023), so measurement and computational errors can be obtained. Another factor in obtaining these errors is the sampling locations on the river at these stations. It is possible that the sampling is only done once in the river station and it matters where in the width of the river the sampling is done, that is, it could be near the river banks or in the center of the river width. Whichever is true, the flow velocity obtained does not entirely reflect the flow velocity of the river at that station since the velocity varies from its position especially since the Pasig River is not a straight flow river. Provided that the obtained flow rate (actual data) is true and correct then the parameter estimation results in Fig. 13 will provide us insights on the best possible true value of the cross-sectional area on the three Stations. Finally, using the estimated values of the flow rates, the approximated values of the TDS concentration will now be very close to the actual values as seen in Figs. 14 and 15. Unfortunately, July of 2019 did not generate an estimate with close to zero error even though it already generated a negative flow rate in the parameter estimation, see Fig. 13C. The relative errors shown in Table 4 are for the TDS concentration using the estimated flow rate. We observe that the errors have been significantly reduced compared to the previous relative error in Table 2. This implies that the parameter estimation captures the true values of the flow rates.

Figure 14 (A–C) Comparison of the approximated TDS concentrations using the model and estimated flow rates at the Guadalupe station to the actual data.

Figure 15 (A–C) Comparison of the approximated TDS concentration using the model and estimated flow rates at the Lambingan station to the actual data.

Table 4 Relative errors between the simulated value using the estimated flow rate and actual data for each month at Lambingan and Guadalupe stations (NA–not available).

Month	Relative error	
	Guadalupe	Lambingan	
	2018	2019	2021	2018	2019	2021	
January	1.57e−02	1.29e−02	0.0187	1.00e−02	1.93e−02	0.0180	
February	3.19e−02	4.39e−02	0.0182	4.09e−03	2.88e−02	0.0033	
March	4.41e−02	6.11e−03	0.0039	1.61e−02	1.42e−02	0.0064	
April	3.31e−02	1.32e−02	0.0231	5.16e−03	4.94e−04	0.0072	
May	6.11e−06	3.33e−05	0.0041	4.42e−05	2.93e−05	0.0089	
June	3.76e−03	3.99e−04	0.0056	2.43e−02	1.40e−04	0.0023	
July	3.10e−02	4.67e−01	0.0140	8.45e−03	7.35e−01	0.0116	
August	3.55e−02	1.66e−02	0.0148	1.43e−02	1.67e−02	0.0357	
September	1.95e−02	1.99e−03	0.0199	1.60e−02	2.90e−03	0.0117	
October	3.06e−02	1.12e−02	0.0039	1.152e−02	7.98e−03	0.0025	
November	5.37e−02	2.54e−02	0.0055	7.56e−04	8.35e−03	0.0330	
December	1.01e−01	NA	0.0070	1.80e−01	NA	0.0080	

Note that one of the main goals of mathematical modeling is to comprehend real-world situations and phenomena in order to find possible solutions and make predictions. In the case of water quality models, predictions of water quality constituents are an important water management factor. This is because one will be able to act swiftly and avoid disastrous effects on water resources if inconsistency is observed. In this work, we obtained a close prediction of the TDS concentration along the Pasig River in the year 2022.

Due to the lack of available data from Pasig River Coordinating and Management Office (PRCMO) (2023) we were not able to include the year 2022 in our simulation results and parameter estimation. However, the lack of data presents an opportunity for us to use the model to predict TDS concentrations in the Pasig River in 2022. In 2021, we were able to estimate the values of the flow rates along the Pasig River. Assuming that the same phenomena or situation will happen in 2022, then the flow rates estimated in 2021 will be used as our predictor for the TDS concentration of 2022. Most of the parameter values in 2022 are available except for the flow rates. Thus, in our simulation, we will use data from 2022 but the flow rates will be from 2021. Any other lacking parameter values in 2022 will be supplied with the available data in 2021 of the same month. With this, Fig. 16 shows the approximated TDS concentration of 2022 in the Pasig River. The relative errors of the predicted TDS concentration for each month in 2022 are given in Table 5.

Figure 16 Comparison of the approximated TDS concentration in 2022 using the model and estimated flow rates from 2021 at the two stations to the actual data.

Table 5 Relative errors between the predicted value and actual data in 2022 for each month at Lambingan and Guadalupe stations.

Month	Relative error	
Guadalupe 2022	Lambingan 2022	
January	0.189	0.150	
February	0.111	0.253	
March	0.142	0.205	
April	0.076	0.587	
May	0.364	0.050	
June	0.119	0.237	
July	0.129	0.195	
August	10.288	14.259	
September	0.064	0.480	
October	0.382	0.508	
November	0.327	0.137	
December	0.805	0.939	

Figure 16 and Table 5 show that, in most months of 2022, the approximated TDS concentration is very close to the actual value. We observed that the relative errors are relatively small for which the majority is less than 25 %, indicating that we have a quite good prediction. The months of August and September, however, do not follow the results of the other months. This implies that there are other factors involved in the months of August and September that resulted in the irregularity of the results. For one, the boundary condition is very crucial to the results in these months since these are the only months whose boundary condition is above 1. This is actually interesting because although the boundary condition is greater than 1, the downstream values are less than 0.5 for the month of August. For the given values of the flow rate, the simulation were not able to catch the results close to the actual values. In situations like this, we reiterate the other factors involved discussed in the preliminary results. Nonetheless, we obtained a close approximation for most months in 2022.

Conclusions

This article presents the application of a water quality model developed by Yeh et al. (2005) and Zhang et al. (2008) to the Pasig River. The model can simulate either sediment transport, reactive water quality constituent transport, or both simultaneously. The water quality constituents are classified as mobile or immobile and the reaction is equilibrium from and to the mobile and immobile water quality constituents. With this, an analytical and numerical solution of the 1D advection-dispersion-reaction model describing the transport of water quality constituents in a river or stream are presented and compared to validate the considered model.

The total dissolved solids (TDS) have been considered in this case study of Pasig river. The simulation was performed using a fully implicit scheme and the finite element method. The simulation result shows an excellent approximation of the concentration of TDS in two stations along the Pasig River. The simulated TDS also has an accepted value for the TDS (Boyd, 2020) except in May, June, and/or July. The results of the simulation and the parameter estimation during May, June, and July demonstrate an unusual behavior not just in the TDS concentration but also in the parameters Q and A. The TDS concentrations during these months are rather larger than the usual values. These values are the results of a smaller cross-sectional area which is the effect of hotter weather conditions during the dry season. Hence, it is important to emphasize that the dry season (where the identified months belong) has affected the TDS concentration greatly. The sources are also an important factor to consider here. If the boundary condition is large, then the concentrations in Guadalupe and Lambingan will most likely have larger values as well. It is good to note that although there is a presence of an extreme value, the model still was able to consider it and provided an excellent approximation of its value.

The sensitivity analysis shows that the parameter Q is more sensitive than the parameter Kx. This indicates that the flow rate has a greater effect on the values of the TDS during simulation than the dispersion coefficient. With this, parameter estimation is performed to obtain precise values of the flow rate Q. The results of the parameter estimation for the flow rate also gave us insights into the true value of the cross-sectional area of the river on the two stations. We observed that the flow rate values are much larger than the computed values, indicating that the cross-sectional area we have been using is a bit different than the exact one. One more interesting result is the usage of the estimated parameter Q in 2021 in predicting the TDS concentration for the year 2022, provided that the same phenomena will happen in the next year. The predictions are very close to the actual values.

The decline in water quality can be measured using a wide range of parameters. The TDS parameter is one that is crucial. An increase in TDS beyond acceptable threshold can have significant impacts on municipal, industrial, and agricultural use of water (Sherrard, Moore & Dillaha, 1987). Water containing no dissolved solids will not support aquatic life. On the other hand, extreme amounts of dissolved solids may not be fit for many human uses and may also be damaging to plants and organisms that depend on fresh water. In most definitions, the total dissolved solids limit is 0.5−1.0 g/L. In excess of these, the river water can be classified as either brackish ( 1.0−10.0 g/L) or saline ( >10.0 g/L) water.

In DENR Administrative Order (DAO) No. 2016-08, no guideline is declared about the total dissolved solids (Pasig River Coordinating and Management Office (PRCMO), 2023). However, because of its influence on other water quality parameters, obtaining its measurement is necessary (Pasig River Coordinating and Management Office (PRCMO), 2023). In the current study, salinization (or increase in TDS concentration) occurs during the dry season for some possible reasons stated in the preliminary results. Fortunately, most of the months have TDS concentrations within the threshold. However, one should consistently and continuously monitor the concentration since an increase of TDS in the freshwater ecosystem beyond stipulated limits is an environmental issue of global concern. If the increase in TDS concentration continues even outside of the dry season, researchers must act swiftly towards preventing an excess beyond the limit; control measures and legislation must be put in place for its regulation. Identification of the cause, whether natural or man-made, is significantly important. Moreover, analysis and prediction is equally important as this research is trying to do. This study can be a stepping stone to further and improve the study of water quality in the Pasig River. With this, an intensive, highly spatial and temporal resolution, and detailed data collection processes, modeling, management and practices must be done on the Pasig River to properly monitor not just the TDS concentration but also other water quality parameters.

For future research and to maximize the use of the model, we suggest to simultaneously look at the distribution of the TDS and other important water quality parameters that it interacts with. We would also like to thoroughly include the influence of other hydrodynamic process like re-suspension and settling velocities, bottom sediments monitoring, tidal effects and more. Furthermore, given sufficient data, we suggest studying the application of the model to the dissolved oxygen (DO) and biological oxygen demand (BOD) in the Pasig river. The model can identify factors that greatly affect the concentration of the DO and BOD and can identify potential threats in the future. Lastly, due to the proximity of the Pasig river to the sea, there is a possibility that the water velocity reverses because of the tides and so, looking at this analysis is also interesting.

Appendix: the pasig river data

In this section, we give the flow velocities and TDS concentrations in the Pasig River stations and tributaries. The data is provided by the Pasig River Coordinating and Management Council upon our request (Pasig River Coordinating and Management Office (PRCMO), 2023). Note: Bold values are missing values from the PRUMS report, these values are either taken from the same month of the previous year or are taken from the preceding month.

For the data, the Pasig River Coordinating and Management Office (Pasig River Coordinating and Management Office (PRCMO), 2023), Department of Environmental and Natural Resources (DENR), National Capital Region, Philippines.

Additional Information and Declarations

Competing Interests

Author Contributions

Data Availability

The authors declare that they have no competing interests.

Crisanto L. Abas conceived and designed the experiments, performed the experiments, analyzed the data, prepared figures and/or tables, authored or reviewed drafts of the article, and approved the final draft.

Arrianne Crystal Velasco conceived and designed the experiments, analyzed the data, authored or reviewed drafts of the article, and approved the final draft.

Carlene Arceo conceived and designed the experiments, analyzed the data, authored or reviewed drafts of the article, and approved the final draft.

The following information was supplied regarding data availability:

The code is available at GitHub and Zenodo:

- https://github.com/CrisLorde/Numerical-resolution-of-a-water-quality-model-and-its-application-to-the-Pasig-River/tree/v1

- CrisLorde. (2024). CrisLorde/Numerical-resolution-of-a-water-quality-model-and-its-application-to-the-Pasig-River: Application of Reaction-based Water Quality Model to the Pasig River (Version v1). Zenodo. https://doi.org/10.5281/zenodo.11238736.

The third party Pasig River data is available upon request from the Pasig River Coordinating and Management Office: Jacqueline A. Caancan, CESO II, Regional Executive Director, DENR-NCR, and Concurrent Executive Director, PRCMO, prcmo.wqwm@gmail.com or denrncrored@gmail.com. The authors received permission to use the data with appropriate citations but not to share it.

The code for the sensitivity analysis is based on the code from Susan Christine Massey of Arizona State University at GitHub: https://github.com/scmassey/model-sensitivity-analysis.

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
