# Peer review of "Application of a reaction-based water quality model to the total dissolved solids concentration of the Pasig River"

_PeerJ, doi:10.7717/peerj.18076_

## Round 0.1 · original submission · Major Revisions

Dear authors: Thank you for submitting this article to PeerJ. I am sending back to you for significant revisions prior to sending out for formal review. I and my colleagues are quite familiar with the models referenced here (Fang et al. 2003; Yeh 2006; Zhang et al. 2008). Most of the material in the sections "Development of the Model" and "Methodology" is not novel and does not warrant repeating at the level of detail given here. Some general discussion of the model formulation is appropriate here, but as written the model development and equations gives the impression to the reader that this is new model development when in fact citing the key papers listed above would be sufficient. The modeling approach used here is in fact sophisticated compared to many river water quality modeling studies, and it would be useful to point out how this modeling framework differs from widely used water quality models (e.g., Qual2K) and why the additional sophistication is needed in the case studied. You do have a brief description of this (lines 68-88) which could be expanded profitably. The literature review should include such widely used models and include more recent publications (most cited are prior to 2010). However, equations and detailed model formulations that are available in the primary cited references should not be repeated here. Aspects that are specific to your case study (e.g. the specific reaction network used for this set of simulations) are appropriate to include.

I request that you significantly modify the manuscript to decrease the focus on model development (which is not novel) and increase the focus on the application study (which is novel). After resubmission I will be happy to send the paper out for peer review.

---

## Round 0.2 · Minor Revisions

Thank you for addressing the editor concerns and restructuring the paper to focus on application of the model.

The paper has now been sent out for peer review and three reviews were received. In my estimation the review comments require only minor revision and clarification.

·

Basic reporting

ok

Experimental design

good

Validity of the findings

ok

Additional comments

good work

Reviewer 2 ·

Basic reporting

- I don´t have the ability to evaluate english;
- The citation/ reference relation was not evaluated;
- Need to insert in the text:
- Input data of flow and TDS for tributaries and main river (with units);
- Units for flow in graphs and units for all variables in tables;
- Maintain the same scale in y-coordinate (makes it easier to identify the longitudinal variation of SDT).

Experimental design

The paper presents the modeling of TDS concentration and not water quality (I suggest adjusting the title); with the current title, it appears the paper focuses on modeling BOD, DO, nutrients, etc.).

The paper focused on the TDS calibration in the mobile water phase! Were the bottom sediments monitored and used as input data in the simulations (chemicals sorbed on sediment) (in g/kg or g/g ?)? Were bottom resuspension velocities assumed in the rainy season and sedimentation in the dry season? As?
Due to the transport and dilution capacity, it was already expected that the flow rate Q would be more sensitive than the dispersion or diffusion coefficient; to better understand the influence of flow rate on TDS concentration, it is important to include an item about hydraulics and hydrodynamics of the watercourse (historical flow series, velocity water, transversal geometry, tidal effect, influence of hydraulic structures!)

Finally, we suggest adding, as a prognosis, the rapid cumulative impacts assessment (support capacity) on the water quality of this river (is there legislation with maximum and minimum limits for parameters (TDS, in this case) (?)

Validity of the findings

We suggest adding, as a prognosis, the rapid cumulative impacts assessment (support capacity) on the water quality of this river (is there legislation with maximum and minimum limits for parameters (TDS, in this case) (?)

Annotated reviews are not available for download in order to protect the identity of reviewers who chose to remain anonymous.

·

Basic reporting

Basically, you are solving the Transport Equation to estimate TDS, but you must mention how the hydrodynamic is estimated (own code, model). First, the hydrodynamic is determined, and then the Transport Equation is applied. You have to mention how this is coupled.

Experimental design

You are giving an exact solution to validate the numerical method applied and then use it in the river. You have to give relative errors with respect the exact solution, and other models have modules to sediment transport, why your model is better?

Validity of the findings

Some concentrations are far from the numerical estimation (again, the estimated errors are necessary). You have to explain why some points are far from the mean concentrations (months).

Additional comments

The work is well presented, and by explaining it in detail, the above can be considered.

---

## Round 0.3 · accepted · Accept

Thank you for the modifications to the manuscript made in both revisions. In my opinion, the manuscript is now acceptable for publication in PeerJ.